# Gradient based Causal Discovery with Diffusion Model

## Abstract

Causal discovery from observational data is an important problem in many applied sciences. Incorporating a recently proposed smooth characterization of acyclicity, gradient-based causal discovery approaches search for a Directed Acyclic Graph (DAG) by optimizing various neural models. Although they show some inspiring results given certain assumptions satisfied, their capability of modeling complex nonlinear causal generative functions is still unsatisfactory. Motivated by recent advances in deep generative models, we propose to use diffusion models for causal discovery, and search for the DAG under continuous optimization frameworks. The underlying nonlinear causal generative process is modeled with diffusion process, and with flexible parameter configurations, it has the ability to represent various functions, and the proposed causal discovery approach are able to generate graphs with satisfactory accuracy on observational data generated by either linear or nonlinear causal models. This is evidenced by empirical results on both synthetic and real data.

## 1 Introduction

Causal discovery from observational data is a research field that has received widespread attention in recent years (Pearl, 2009; Spirtes et al., 2001; Peters et al., 2017) which aims to automatically learn the causal relationship (directed acyclic graph, DAG) between variables from non-experimental data. To tackle the NP-hard structure learning problem(Chickering et al., 2004), constraint-based methods uses conditional independence tests to locate causal edges between variable pairs admissible to the given dependence measurements (Spirtes et al., 2001; Zhang, 2008). On the other hand, score-based methods deploy a score to measure the fitness of a causal graph over data, with a suitable search procedure such as greedy search (Chickering, 2002) and dynamical programming (Silander & Myllymaki, 2012). Bayesian score criteria is usually adopted, including Bayesian information criteria (BIC, (Maxwell Chickering & Heckerman, 1997)), Bayesian Gaussian equivalent (BGe) score (Kuipers et al., 2014) and minimum description length (MDL, (Bouckaert, 1993)) with consistency and equivalence property maintained. A common challenge is that the complexity of the search space is superexponential with the number of variables, which poses a big challenge to the optimization procedure.

Recently, an acyclic constraint of the adjacency matrix that is a continuous function is proposed (Zheng et al., 2018). Transforming the traditional combinatorial nature of the acyclicity to a continuous optimizable one, it triggers a series of differentiable causal discovery approaches, considering various model settings (Zheng et al., 2018; Lee et al., 2019). Yu et al. (2019) incorporates the acyclicity constriants with the framework of variational autoencoders (Kingma, 2013) for causal discovery, and Zhu et al. (2019) uses reinforcement learning (Kaelbling et al., 1996) as the search algorithm. Although the nonlinearity of the optimization problem often makes the solutions only local optimal, empirical results show certain closeness to that obtained by performing exhaustive combinatorial search (Zhu et al., 2019).

Nonetheless, most deep learning (generative) model based approaches tie to a specific functional model class, the generalized linear causal models. The model assumptions already set limitations on the representation capability, and may fail to model observational data distributions when they follow a complicated underlying generative process. An example is the post-nonlieanr causal model (Zhang & Hyvärinen, 2009), where a post-nonlinear transformation directly follows an additive noise model

(Peters et al., 2014) due to reasons such as sensor bias. Admitting a more general nonlinear causal models is thus helpful for wider applicability of differentiable algorithms. Meanwhile, learning a DAG via optimizing the evidence lower bound of variational autoencoders (Yu et al., 2019), reward function of reinforcement learning (Zhu et al., 2019), or the score function with the nonparametric DAG constraint (Zheng et al., 2020) is known to suffer from certain unstability due to the richness of the search space, presenting possible reliability issues on the results.

Motivated by remarkable success of deep generative models, we propose to use diffusion model (Croitoru et al., 2023; Sohl-Dickstein et al., 2015) for causal discovery. The proposed approach has a natural fit for the following nonlinear generative model

$$X = g((I - A)^{-1}f(Z)) \tag{1}$$

where $X \in \mathbb{R}^n$ and $Z \in \mathbb{R}^n$ are the $n$-dimensional observed variable and mutually-independent noise, respectively, and $f$ and $g$ are some nonlinear functions. $A$ is the $n$-square weighted adjacency matrix. By the diffusion process represented by ordinary or stochastic differentiable equation (ODE or SDE) (Ho et al., 2020; Song et al., 2020b; Karras et al., 2022), the model has the flexibility of capturing the complex nonlinear causal functions by various denoising variance sequence, and the DAG learning is via denoising score matching. Experiments on both synthetic and real world data show that our approach has improved performance on several settings including models with nonlinearity.

## 2 RELATED WORK

Constraint-based causal discovery methods initiate their process by leveraging conditional independence tests to outline a causal skeleton. Edge orientation approaches are then applied to determine DAGs up to Markov equivalence class. Examples include PC algorithm and SGS algorithm (Spirtes et al., 2001) and several extensions when kernel based conditional independence measurements are used (Zhang et al., 2011; Sun et al., 2007). Nonetheless, the accuracy of such methods heavily relies on the reliability of the independence tests, and resolving possible conflicting constraints caused by multiple independently-conducted tests remains substantially challenging (Tsamardinos et al., 2012). The DAG identification framework with experimental data available is also considered (Hyttinen et al., 2013).

Score-based causal discovery mainly uses a given score to measure the quality of the causal graph, coupled with algorithms to search for the optimal graph (Rolland et al., 2022). Scores include BIC (Maxwell Chickering & Heckerman, 1997) and MDL (Bouckaert, 1993), and the generalized kernel-based scoring function (Huang et al., 2018). By incorporating additional assumptions about the data distribution and/or underlying generative functions, scores based on well-defined functional causal models are proposed. These methods differ from constraint-based approaches, which often presume faithfulness and are limited to identifying the Markov equivalence class. Given assumptions satisfied, these functional causal model-based methods can effectively distinguish between distinct directed acyclic graphs (DAGs) within the same equivalence class, with applications covering linear non-Gaussian model (Shimizu et al., 2006), the nonlinear causal models with additive noise (Hoyer et al., 2008; Peters et al., 2014), and the post-nonlinear causal model (Zhang & Hyvärinen, 2009) which accommodates a post-nonlinear transformations after the nonlinear causal functions. Recently, diffusion model based estimation of the Hessian matrix of the additive noise model has also been investigated (Sanchez et al., 2022).

Differentiable causal discovery is based on a recent smooth characterization of acyclicity constraint (Zheng et al., 2018), which transforms the combinatorial DAG search problem to be continuously optimizable. This enables searching for a DAG with gradient-based optimizations in an end-to-end way (Geffner et al., 2022), using various black-box solvers. Different searching approaches have been proposed, considering linear models (NOTEARS, (Zheng et al., 2018)), nonlinear variational autoencoder-based models (DAG-GNN (Yu et al., 2019)) and reinforcement learning-based models (RL-BIC, (Zhu et al., 2019)). Wei et al. (2020) presents a theoretical perspective that the edge missing in linear model equals to a non-satisfaction of KKT condition in optimization, motivating a local search algorithm. Lee et al. (2019) and Lachapelle et al. (2019) utilize polynomial regression and neural networks to deal with nonlinear causal relationships with flexibility to represent conditional distributions, respectively. Zheng et al. (2020) reconsiders the algebraic smooth constraint of acyclicity in the perspective of nonparametric structural equation models, via leveraging sparsity

characterized by partial derivatives. Causal discovery via graph autoencoder (Ng et al., 2019) and at latent conceptual level (Yang et al., 2021) are also studied.

Diffusion models are recently shown to be with superior performance on various application domains upon content generations (Cao et al., 2024). Unlike generative adversarial networks (Goodfellow et al., 2020) or variational authencoders (Kingma, 2013), it essentially consists of two intricately linked processes: a predefined forward process and a corresponding reverse process. The forward process transforms the data into a simple prior distribution, typically a Gaussian, while the reverse process leverages a trained neural network to progressively reverse the effects of the forward process following stochastic differential equations. The neural network for the denoising process (usually Markovian) is trained under the denoising score-matching function (Song et al., 2020b). The capability of the model on capturing complex nonlinear generative process is from the flexibility of configuring the denoising variance sequences, with remarkable success on bio-informatics (Xu et al., 2022), natural language processing (Li et al., 2022) and computer vision (Croitoru et al., 2023). The property is possibly beneficial for representation learning (Abstreiter et al., 2021).

## 3 STRUCTURE LEARNING VIA DIFFUSION MODEL

In this section, we propose our diffusion model based causal discovery method. The basic idea is to use the deep diffusion process to model nonlinear data generative mechanisms, incorporating the designed causal layer with an adjacency matrix being an learnable element. To start with, we introduce the denoised diffusion probabilistic model (DDPM), which is a discrete approximation (Markov process) of the continuous diffusion process represented by stochastic differentiation equation (Ho et al., 2020).

### 3.1 DENOISED DIFFUSION PROBABILISTIC MODEL

In diffusion models, the generative process unfolds over a direction of time. It represents a sequence of data distributions that characterize the model's gradual transformation from the initial data distribution to a target noise distribution. Denote the initial state as $Y_0$, sampled from the data distribution. Noise is incrementally added at each subsequent time step to push the distribution to gradually converge towards a known prior state $Y_t$ (normally following a Gaussian distribution). The evolution of the data distribution is through the fine-gained sequence of intermediate states, each corresponding to a specific time point within the process. By varying the setting of noise addition, the diffusion model is able to capture a variety of underlying generative mechanisms. To be mathematically precise (Ho et al., 2020), we denote the forward process $\mathcal{F}$ and the reverse process $\mathcal{R}$, and the diffusion and reverse process then can be written as

$$\mathcal{F}(Y_0, \Gamma) = \mathcal{F}_T(Y_{T-1}, \gamma_T) \circ \cdots \circ \mathcal{F}_t(Y_{t-1}, \gamma_t) \circ \cdots \circ \mathcal{F}_1(Y_0, \gamma_1),$$
$$\mathcal{R}(Y_T, \Gamma) = \mathcal{R}_T(Y_1, \gamma_1) \circ \cdots \circ \mathcal{R}_t(Y_t, \gamma_{t-1}) \circ \cdots \circ \mathcal{R}_T(Y_T, \gamma_T), \quad (2)$$

where the transition kernels (function) at time step $t$ are $\mathcal{F}_t$ for forward process and $\mathcal{R}_t$ for backward process respectively, and $\Gamma = \{\gamma_i\}_{i=1}^T$ are a sequence of variance that controls the diffusion process. The possible choices of variance sequences are linear decreasing or cosine like schedules (Song et al., 2020b). Following the DDPM, we define $\mathcal{N}$ the multivariate Gaussian distribution and $q$ being some distribution function, $\mathbf{I}$ the standard identity matrix. Then the forward transition kernel is

$$\mathcal{F}_t(Y_{t-1}, \gamma_t) = q(Y_t|Y_{t-1}) = \mathcal{N}(Y_t; \sqrt{1-\gamma_t}Y_{t-1}, \gamma_t\mathbf{I}). \quad (3)$$

The concatenation of the transition kernels builds up the forward diffusion process, which transits the initial state of the variable $Y_0$ to be a predefined prior distribution $Y_T$. Correspondingly, the reverse process consists of learnable Gaussian kernels parameterized by $\theta$ as

$$\mathcal{R}_t(Y_t, \Sigma_{t-1}) = p_\theta(Y_{t-1}|Y_t) = \mathcal{N}(Y_{t-1}; \mu_\theta(Y_t, t), \Sigma_\theta(Y_t, t)), \quad (4)$$

$\mu_\theta$ and $\Sigma_\theta$ are parameterized Gaussian kernel determined by the reverse sequence distribution $p_\theta$. It is shown that DDPM resembles to approximate the observed data distribution by the reconstructed probability distribution (Song et al., 2020b) $p_\theta(Y_0) = \int p_\theta(Y_{0:T})dY_{1:T}$.

The training of the diffusion model is by minimizing the following variational bound on the negative log likelihood, defined over KL-Divergence operators as

$$\mathbb{E}[-\log p_\theta(Y_0)] \leq \mathbb{E}_q[\mathcal{D}(q(Y_T|Y_0)\|p(Y_T))$$
$$+ \Sigma_{t>1}\mathcal{D}(q(Y_{T-1}|Y_t, Y_0)\|p_\theta(Y_{T-1}|Y_t)) - \log p_\theta(Y_0|Y_1)] \quad , \quad (5)$$

$\mathcal{D}(\cdot\|\cdot)$ is the KL divergence of two distributions. The first and third term of the RHS of (5) are the prior and reconstruction losses. The middle of the RHS of (5) denotes the divergence sum between the posterior of the forward and reverse distributions over each time step $t$. Conditioning on some simplifications, we can rewrite the posterior distribution as

$$q(Y_{t-1}|Y_t, Y_0) = \mathcal{N}(Y_{t-1}; \tilde{\mu}_t(Y_t, Y_0), \tilde{\gamma}_t \mathbf{I}), \tag{6}$$

where $\tilde{\gamma}_t$ is a function of $\gamma_t$. Then one can reparameterize the KL divergence to be a $l_2$-loss between the two mean coefficients

$$\mathcal{D}(q(Y_{T-1}|Y_t, Y_0)\|p_\theta(Y_{T-1}|Y_t)) = \mathbb{E}_q[\frac{1}{2\gamma_t^2}\|\tilde{\mu}_t(Y_t, Y_0) - \mu_\theta(Y_t, t)\|^2] + \text{const}. \tag{7}$$

The optimization is then done by solving the denoised score matching (Ho et al., 2020) process whose objective function sums over the single-step loss described by (7), and furthur simplifications can be made to improve the training strategy (Cao et al., 2024).

## 3.2 REPRESENTING STRUCTURAL CAUSAL MODEL BY DIFFUSION PROCESS

In this section, we describe the nonlinear structural causal models (SCM). Recall (1) introduced in section 1, we follow the settings and present a linear SCM as

$$X = AX + Z. \tag{8}$$

$A$ is the weighted adjacency matrix associated with a directed acyclic graph, with $A_{ij}$ non-zero indicating a causal edge from node $i$ to node $j$. The acyclicity of $A$ corresponds to the fact that $A$ can be permuted to be strictly upper (or lower) triangular. NOTEARS (Zheng et al., 2018) generates the adjacency matrix by searching for a solution that results in minimum linear reconstruction error over data, subject to $A$ being acyclic, under a smooth characterization of acyclicity. Assuming the $Z$ to be non-Gaussian, the model (8) becomes the linear non-Gaussian acyclic model, where graph learning can be solved by independent component analysis (Shimizu et al., 2006), because we can transform the problem as

$$X = (I - A)^{-1}Z \tag{9}$$

assuming invertibility. The uniqueness of the solution is based on non-Gaussianity of the independent source (noise) variables (Hoyer et al., 2008). In this paper, we consider a more general nonlinear extension of (8)

$$X = g((I - A)^{-1}f(Z)). \tag{10}$$

where $f$ and $g$ are variable wise (multidimensional) functions. The noise passes through a nonlinear transformation, and then a mixing process, a post-nonlinear transformation to be the observed variables $X$ with causal structures. When $f$ and $g$ are invertible, we can write the causal model as

$$g^{-1}(X) = Ag^{-1}(X) + f(Z). \tag{11}$$

Unlike the work (Yu et al., 2019) in which the $f$ is suggested to be an identity function, we here put $f$ to be some nonlinear function, admitting a richer class of joint distributions. Then consider a diffusion process that simulates the aforementioned structural causal model (11) as

$$X \rightarrow \text{diffusion} \rightarrow (I - A) \rightarrow \text{diffusion} \rightarrow Z \rightarrow \underbrace{\text{reverse}}_{f} \rightarrow (I - A)^{-1} \rightarrow \underbrace{\text{reverse}}_{g} \rightarrow \bar{X}, \tag{12}$$

where the forward diffusion process simulates the model that can be a reformulation of equation (10)

$$Z = f^{-1}((I - A)g^{-1}(X)). \tag{13}$$

Then we formally define the diffusion model for causal discovery, by reformulating the $T$-step diffusion process by (2) with graph associated weighted adjacency matrix $A$ in the function

$$\begin{aligned}
\mathcal{F}_{\mathcal{G}}(X, A, \Gamma) &= \mathcal{F}_T(X_{T-1}, \gamma_T) \circ \cdots \circ (I - A) \circ \cdots \circ \mathcal{F}_1(X_0, \gamma_1), \\
\mathcal{R}_{\mathcal{G}}(Z, A, \Gamma) &= \mathcal{R}_1(\bar{X}_1, \gamma_1) \circ \cdots \circ (I - A)^{-1} \circ \cdots \circ \mathcal{R}_T(Z, \gamma_T).
\end{aligned} \tag{14}$$

$X_0 = X$ is the initial state of variable $X$ enrolled in the diffusion process, and $\bar{X}_t$ is the reconstructed $X_t$ at time stamp $t$. Under this formulation, the underlying generative functions of the

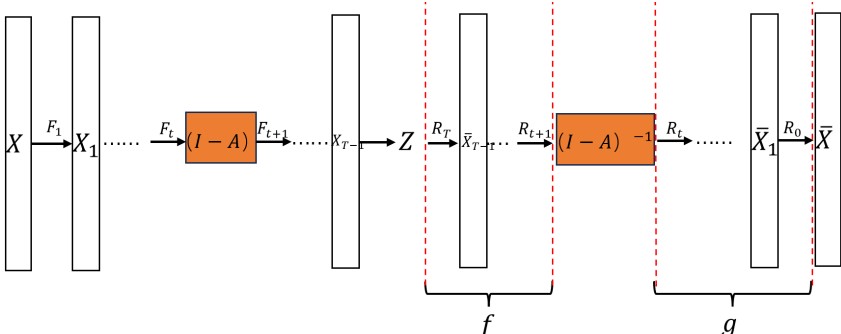

Figure 1: Illustration of the diffusion based causal modeling.

causal model (10) can be modeled by a concatenation of transition kernels in the forward or backward process. An example is

$$p(f(Z)) = \prod_{i=t}^{T} \mathcal{R}_i(\bar{X}_i, \gamma_i) \tag{15}$$

where $Z$ (the mutually-independent multidimensional noise variables) is also represented by the $T$-step transformation of $X$ as $\bar{X}_T$. By different parameterization of the transition kernels including the variance sequence $\Gamma$, one may get a rich class of functions that the diffusion process can model. Some theoretical analysis shows that the discrete diffusion process is an approximation of the stochastic differential equation (Song et al., 2020b), which simulates a Brownian movement transforming the original variable to noise. In the next section, we will present a graph searching strategy by incorporating the differentiable acyclicity constraint on $A$.

### 3.3 Optimizations

In this section, we discuss the core elements of the model based optimization for graph search, basically consisting of objective functions, acyclicity constraints and the acceleration strategies when efficiency is also of consideration.

**Acyclicity constraints.** The acyclicity of the adjacency matrix $A$ should be enforced in the graph searching procedure. Approaches such as greedy equivalence search (Chickering, 2002) check explicitly for the satisfaction of acylic constraint when each edge operation is conducted, e.g., edge addition, which is of high complexity when the number of node is large. In this work, we optimize the matrix $A$ as a whole under a continuous framework, admitting simultaneous change of multiple entries in a single round, and we thus adopt the recently proposed smooth characterization of the acyclicity (Zheng et al., 2018). A directed graph associated weighted adjacency matrix $A$ is acyclic if and only if

$$h(A) = \text{trace}(e^{A \odot A}) - n = 0, \tag{16}$$

where $\odot$ is the Hadamard product and $e^A$ is the matrix exponential which can be interpreted as a Taylor expansion of power series that ties to the reweighed number of closed walks in the matrix.

**Objective function.** Recall the modified diffusion process with the graph inserted, $\mathcal{F}_\mathcal{G}(X, A, \Gamma)$ and $\mathcal{R}_\mathcal{G}(Z, A, \Gamma)$ in (14). The objective function is the evidence lower bound described in (5), with single-step loss estimation approach as (7). Given an set of observational data of some variable $X$, denote the loss function over the samples as $\mathcal{L}(A, \Theta)$ where $\Theta$ is a set of all editable parameters needed to determine the diffusion process. Incorporating an acyclicity constraint, the optimization problem becomes

$$\min_{A,\Theta} \mathcal{L}(A, \Theta), \text{ subject to } h(A) = 0. \tag{17}$$

We can transform the constrained problem to an unconstrained one using the Lagrange multiplier approach as

$$\min_{A,\Theta} \mathcal{L}(A, \Theta) + \lambda h(A). \tag{18}$$

---

**Algorithm 1** Diffusion model based causal discovery

---

1: Initialize $(A_0, \Theta_0, \lambda_1)$, progressive parameters $c, \omega, \eta$;
2: **for** $i = 1, 2, ...$ **do**
3:    $(A_i, \Theta_i) = \arg\min_{A,\Theta} \mathcal{L}(A, \Theta) + \lambda_i h(A)$ ;
4:    $\lambda_{i+1} = \lambda_i + ch(A_i)$;
5:    **if** $h(A_i) > \omega h(A_{i-1})$ **then**
6:      $c = \eta c$;
7:    **end if**
8:    Break if converged;
9: **end for**
10: Return the estimated adjacency matrix.

---

The problems (17) and (18) are basically equivalent when some conditions are satisfied. To be mathematically precise, we present the following proposition which can be proved under the logic of showing some equivalence between the minimizers of the two problems following Zhu et al. (2019). A proof is provided in Appendix A.

**Proposition 1.** Let $h_* > 0$ be the minimum of $h(A)$ over all direct cyclic graphs, i.e., $h_* = \inf_{A \notin \mathrm{DAGs}} h(A)$. Assume that the loss function $\mathcal{L}(A, \Theta)$ is bounded with $\mathcal{L}_l = \inf \mathcal{L}(A, \Theta)$ and $\mathcal{L}_u = \sup \mathcal{L}(A, \Theta)$. Then problems (17) and (18) are equivalent if

$$\lambda h_* \geq \mathcal{L}_u - \mathcal{L}_l. \tag{19}$$

The proposition provides a principled guide for the choices of some parameters such as the penalty parameters. Although accurate estimations of the optimal values are non-trivial, one may get a rough guess of these parameters via some approaches. For example, feeding DAGs generated randomly or by linear methods (such as NOTEARS (Zheng et al., 2018)) may help to guess the bounds of the loss function. In fact, the important intuition the proposition carries is that with large enough $\lambda$, the equivalence of the problems are with full guarantee. This motivates us to use the augmented Lagrangian which optimizes a series of subproblems, with their exact solutions converging to a stationary point of the original constrained problem under certain conditions (Bertsekas, 1997). This enables adaptive updating of the penalty parameters to enforce acyclicity, similar to the approach used by Zheng et al. (2018); Yu et al. (2019). Each subproblem (augmented Lagrangian) writes as

$$\min_{A,\Theta} \mathcal{L}(A, \Theta) + \lambda_i h(A) + \frac{c_i}{2} h(A)^2. \tag{20}$$

$c_i$ is some penalty parameters. Parameter updating is performed after sub-problem $i$ resolved and we progress to the state $i + 1$. The solution of the previous problem is used as the initialization point of the next subproblem, so that the optimization procedure enjoys certain consistency. Common non-convex optimization solvers are applied to approximately solve the problems, e.g., RMSprop (Hinton et al., 2012) and Adam (Kingma, 2014), where stochastic gradient descent is normally employed for searching a local minimum. We detail the approach in Algorithm 1.

**Acceleration.** The diffusion models are normally with deep layers compared to encoder-decoder based generative models like variational autoencoders. One important issue is to maintain efficiency under a certain standard of accuracy. The work (Song et al., 2020b) extends DDPM to a continuous stochastic differential equation based framework, and further shows that a probability flow ordinary differential equation, supporting the deterministic diffusion process, shares the same marginal distribution of $X$. We here treat $X$ as a continuous time-dependent variable in the diffusion process.

$$dX = \{k_1(X, t) - \frac{1}{2} k_2(t)^2 \nabla_X \log p_t(X)\} dt, \tag{21}$$

where $k_1(, t)$ is the drift coefficient of $X(t)$, and $k_2$ is some diffusion coefficients, $p_t(X)$ is the marginal distribution. By reparameterizing the ODE-based sampling process, the Denoising Diffusion Implicit Model (Song et al., 2020a) is proposed to accelerate the process. Utilizing a non-Markovian sampling process, it is able to achieve inference using less steps, with applicable mature solvers (Lu et al., 2022). In experimental section, we adopt DDIM for higher efficiency. Depending on the application scenarios, one should be aware of the possibility of applying other SDE or ODE based samplers, e.g., analytical method (Bao et al., 2022) and dynamic programming adjustments (Watson et al., 2021).

Table 1: **Empirical results on linear models with Gaussian noise.**

| nodes | | DAG-Diffusion | PC | CAM | ICA-LiNGAM | DAG-GNN | NOTEARS | NOTEARS-MLP | GAE | DiffAN |
|---|---|---|---|---|---|---|---|---|---|---|
| 10 | TPR | 0.92 ±0.004 | 0.44 ±0.16 | 0.55 ±0.16 | 0.18 ±0.02 | 1 ±0 | 0.89 ±0.05 | 0.94 ±0.12 | 0.87±0.06 | 0.45±0.17 |
| | FDR | 0.04 ±0.06 | 0.46 ±0.17 | 0.52 ±0.16 | 0.86 ±0.02 | 0.03 ±0.03 | 0.03 ±0.01 | 0.05 ±0.10 | 0.19±0.18 | 0.69±0.15 |
| | SHD | 1.3 ±0.5 | 24 ±5 | 7 ±2 | 24 ±6 | 0.5 ±0.4 | 2 ±1.8 | 0.7 ±1.42 | 4 ± 3 | 14±6 |
| | F1 score | 0.93 ±0.03 | 0.42 ±0.19 | 0.46 ±0.19 | 0.05 ±0.01 | 0.98 ±0.02 | 0.92 ±0.05 | 0.94 ±0.11 | 0.83±0.13 | 0.3±0.2 |
| 50 | TPR | 0.85 ±0.05 | 0.8 ±0.05 | 0.62 ±0.05 | 0.19 ±0.03 | 0.95 ±0.02 | 0.98 ±0.01 | 0.92±0.03 | 0.67±0.05 | 0.37±0.02 |
| | FDR | 0.16 ±0.08 | 0.27 ±0.06 | 0.47 ±0.07 | 0.88 ±0.05 | 0.03 ±0.03 | 0.03 ±0.01 | 0.11±0.06 | 0.3±0.08 | 0.85±0.01 |
| | SHD | 15 ±6 | 23 ±5 | 29±6 | 82 ±18 | 4 ±3 | 1 ±0.9 | 9±7 | 30±6 | 106±19 |
| | F1 score | 0.83 ±0.05 | 0.78 ±0.05 | 0.56 ±0.06 | 0.04 ±0.01 | 0.96 ±0.03 | 0.98 ±0.011 | 0.9±0.05 | 0.68±0.05 | 0.11±0.01 |

Table 2: **Empirical results on linear models with non-Gaussian noise.**

| nodes | | DAG-Diffusion | PC | CAM | ICA-LiNGAM | DAG-GNN | NOTEARS | NOTEARS-MLP | GAE | DiffAN |
|---|---|---|---|---|---|---|---|---|---|---|
| 10 | TPR | 0.95 ±0.05 | 0.74 ±0.19 | 0.22±0.13 | 0.02 ±0.05 | 0.98 ±0.03 | 0.97 ±0.05 | 0.89 ±0.15 | 0.73±0.09 | 0.58±0.13 |
| | FDR | 0.12 ±0.12 | 0.38 ±0.118 | 0.8 ±0.09 | 0.97 ±0.04 | 0.02 ±0.02 | 0.03 ±0.02 | 0.10 ±0.14 | 0.26±0.15 | 0.67±0.11 |
| | SHD | 1.5 ±0.5 | 7±3 | 12 ±4 | 12 ±3 | 0.5 ±0.4 | 0.5 ±0.3 | 1.4 ±2.06 | 5±3 | 13±5 |
| | F1 score | 0.9 ±0.04 | 0.66±0.15 | 0.09±0.07 | 0.01±0.01 | 0.98±0.01 | 0.97±0.05 | 0.89±0.14 | 0.73±0.11 | 0.36±0.16 |
| 50 | TPR | 0.82 ±0.07 | 0.82 ±0.1 | 0.39 ±0.05 | 0 ±0 | 0.93 ±0.01 | 0.93 ±0.04 | 0.89±0.05 | 0.63±0.06 | 0.43±0.11 |
| | FDR | 0.08 ±0.07 | 0.3 ±0.06 | 0.7 ±0.05 | 1 ±0 | 0.13 ±0.02 | 0.03±0.01 | 0.15±0.05 | 0.2±0.09 | 0.79±0.09 |
| | SHD | 13 ±5 | 25 ±6 | 53 ±13 | 61 ±5 | 8 ±1 | 4 ±4 | 8±5 | 27±9 | 100 ±34 |
| | F1 score | 0.86±0.05 | 0.75 ±0.08 | 0.23 ±0.06 | 0 ±0 | 0.9±0.01 | 0.95 ±0.04 | 0.86±0.03 | 0.7±0.07 | 0.19±0.14 |

# 4 EXPERIMENTS

In this section, we report empirical results on synthetic and real datasets, comparing performance of our proposed method to both score-based and constraint-based methods, including PC (with Fisher independence test upon threshold on p-value 0.01) (Spirtes et al., 2001), Causal Additive Model (Bühlmann et al., 2014), ICA-LiNGAM (Shimizu et al., 2006), DAG-GNN (Yu et al., 2019) and GraN-DAG (Lachapelle et al., 2019), NOTEARS-MLP (Zheng et al., 2020), GAE (Ng et al., 2019) and DiffAN (Sanchez et al., 2022). The implementations of all these algorithms are from available online source and more details are provided in Appendix C. Unless stated, default settings of the algorithms are used in the experiments. The thresholding method for graph pruning (e.g., abandoning an edge if the estimated (absolute) value of the corresponding entry in the adjacency matrix is less than a threshold) is adopted for ICA-LiNGAM, NOTEARS and DAG-GNN, NOTEARS-MLP and GAE. For CAM, DiffAN and GraN-DAG, the significance testing of covariates is applied and the threshold of p-values for declaration of significance is 0.001.

The proposed diffusion based algorithm named DAG-Diffusion is implemented based on PyTorch implementations of denoised diffusion probabilistic models (Ho et al., 2020). The forward and reverse process are then reformed as described by (14). The reverse process is accelerated by implementing DDIM modules for efficient sampling. More details are reported in Appendix B.

The evaluation metrics include False Discovery Rate (FDR), True Positive Rate (TPR), F1 score and the Structural Hamming Distance (SHD) which is the minimum number of edge operations (including additions, deletions and reversals) needed to convert the estimated graph to the given true DAG. The SHD is a comprehensive measurement considering both the false positives and false negatives, and it is widely used for quantifying the accuracy of causal discovery algorithms. A lower SHD indicates a better estimated graph.

## 4.1 LINEAR MODEL WITH ADDITIVE NOISE

Given a dimension of matrix $n$, we generate the $n$-node graphs using the Erdős-Rényi approach, which randomly samples a topological order and adds directed edges independently. The degree of each node is set to be 2. Given a graph, the associated weighted adjacency matrix $A$ is obtained by assigning the edge weights independently from a uniform distribution on $[0.5, 2]$ with random signs. Standard Gaussian or non-Gaussian noise $Z$ is then sampled to generate the data following the linear model

$$X = AX + Z. \tag{22}$$

For non-Gaussian noise, the distribution is set to be Gumbel distribution with parameter (location,scale) to be $(0, 1)$. Sample size is fixed to be 1000 over all experiments and the sampling procedure is similar to that used by Yu et al. (2019). The identifiability of Gaussian and non-Gaussian models are discussed in several works (Shimizu et al., 2006; Bühlmann et al., 2014). We

Table 3: **Empirical results on nonlinear models with Gaussian noise.**

| nodes | | DAG-Diffusion | PC | CAM | ICA-LiNGAM | DAG-GNN | NOTEARS | NOTEARS-MLP | GAE | DiffAN |
|---|---|---|---|---|---|---|---|---|---|---|
| 10 | TPR | 0.66±0.21 | 0.6±0.16 | 0.59±0.1 | 0.23±0.09 | 0.15±0.12 | 0.15±0.07 | 0.33 ±0.14 | 0.41±0.17 | 0.56±0.18 |
| | FDR | 0.43±0.1 | 0.5±0.09 | 0.28±0.06 | 0.7±0.1 | 0±0 | 0±0 | 0.0 ±0.0 | 0.13±0.16 | 0.3±0.22 |
| | SHD | 8±1 | 10±3 | 6±1 | 11±2 | 10±2 | 10±2 | 7 ±3 | 6±4 | 5±2 |
| | F1 score | 0.59±0.1 | 0.52±0.1 | 0.63±0.06 | 0.15±0.07 | 0.24±0.2 | 0.25±0.13 | 0.48 ±0.16 | 0.55±0.17 | 0.58±0.21 |
| 50 | TPR | 0.62±0.1 | 0.73±0.05 | 0.54±0.1 | 0.26±0.03 | 0.06±0.05 | 0.1±0.05 | 0.51±0.06 | 0.41±0.12 | 0.41±0.04 |
| | FDR | 0.31±0.03 | 0.51±0.04 | 0.34±0.09 | 0.73±0.08 | 0.05±0.02 | 0.05±0.05 | 0.16±0.08 | 0.31±0.16 | 0.56±0.05 |
| | SHD | 32±7 | 47±4 | 29±3 | 55±3 | 53±8 | 51±8 | 23±1 | 42±4 | 36±3 |
| | F1 score | 0.66±0.1 | 0.57±0.03 | 0.58±0.05 | 0.15±0.06 | 0.12±0.1 | 0.17±0.08 | 0.62±0.05 | 0.48±0.06 | 0.36±0.06 |

Table 4: **Empirical results on nonlinear models with non-Gaussian noise.**

| nodes | | DAG-Diffusion | PC | CAM | ICA-LiNGAM | DAG-GNN | NOTEARS | NOTEARS-MLP | GAE | DiffAN |
|---|---|---|---|---|---|---|---|---|---|---|
| 10 | TPR | 0.65±0.17 | 0.63±0.19 | 0.49±0.13 | 0.17±0.13 | 0.1±0.06 | 0.13±0.12 | 0.43 ±0.05 | 0.43±0.13 | 0.5±0.2 |
| | FDR | 0.28±0.1 | 0.47±0.04 | 0.4±0.1 | 0.77±0.1 | 0±0 | 0±0 | 0.05 ±0.07 | 0.46±0.28 | 0.39±0.19 |
| | SHD | 8±1 | 11±3 | 5±1 | 10±2 | 10±2 | 9±3 | 5 ±1 | 9±5 | 5±2 |
| | F1 score | 0.61±0.1 | 0.55±0.12 | 0.51±0.11 | 0.1±0.1 | 0.15±0.11 | 0.22±0.17 | 0.58 ±0.05 | 0.4±0.24 | 0.51±0.25 |
| 50 | TPR | 0.45±0.05 | 0.1±0.02 | 0.54±0.05 | 0.21±0.06 | 0.09±0.05 | 0.08±0.1 | 0.46±0.04 | 0.44±0.21 | 0.57±0.08 |
| | FDR | 0.16±0.1 | 0.57±0.06 | 0.41±0.05 | 0.73±0.09 | 0±0 | 0.1±0.1 | 0.22±0.04 | 0.6±0.18 | 0.24±0.09 |
| | SHD | 25±6 | 50±4 | 30±4 | 51±10 | 45±7 | 46±7 | 29±5 | 70±16 | 26±5 |
| | F1 score | 0.6±0.05 | 0.48±0.05 | 0.56±0.06 | 0.12±0.07 | 0.15±0.09 | 0.14±0.1 | 0.57±0.04 | 0.35±0.16 | 0.65±0.07 |

use a thresholding based pruning method which truncates an estimated edge if the absolute value of the corresponding entry is less than a threshold. The value is determined heuristically: after the algorithm's output, the average of all absolute values of predicted edges divided by 2 is calculated, denoted as $\alpha$. Then the threshold is round$(10\alpha)/10$.

The empirical results on linear Gaussian and non-Gaussian models are reported in Table 1 and Table 2, respectively. The performance of PC, CAM and GAE is reasonable but less satisfactory. This is possibly because these algorithms mostly target nonlinear cases. DiffAN sometimes has the wrong direction estimations. This possibly relates to the identifiability problems of on observational data in linear cases with Gaussian additive noise. ICA-LiNGAM performs poorly among all settings. In linear Gaussian models, it is reasonable since the non-Gaussian assumptions of ICA-LiNGAM are violated. The unsatisfactory performance in non-Gaussian models is possibly because the generative model produces data distributions that are close to the indeterminacy cases of independent component analysis, and ICA-LiNGAM is unable to recover the correct adjacency matrix. DAG-GNN and NOTEARS, NOTEARS-MLP have best performance among all algorithms, showing their capability of dealing with linear models with additive noise, which admits their model assumptions. Our DAG-Diffusion has satisfactory performance in both Gaussian and non-Gaussian cases, showing the possibility of diffusion process to model linear functions under various setting of node sizes (10 and 50).

## 4.2 NONLINEAR MODEL WITH ADDITIVE NOISE

We consider the nonlinear causal relationships with random nonlinear functions. The model is with the following form

$$X = \text{MLP} \circ A(X + 0.5) + Z. \tag{23}$$

The MLP is a multi-layer perception nets with weights randomly assigned. The number of layers is 3 for 10-node case and 2 for 50-node case. The noise $Z$ is with distributions to be either standard Gaussian or non-Gaussian for different model settings, and the sample size of the experiments is fixed to be 1000 among all tests. Since the performance of diffusion model is known to be with some sensitivity to the sample size, we also put down a sample size related study on Appendix D.

Empirical results on nonlinear models are reported in Table 3 and 4. In nonlinear cases, PC has satisfactory performance among all algorithms. The performance of CAM, GAE and NOTEARS-MLP are satisfactory among all methods, which are empirical evidences for their capability for nonlinear modeling. DiffAN works also well on these cases, but its performance is with relatively less stableness considering the variance. ICA-LiNGAM and NOTEARS have descent performance in both Gaussian and non-Gaussian models. A large set of edges is not discovered by them correctly. The performance deterioration is because of the violation of the linear model assumptions. DAG-GNN's experimental results are on par with NOTEARS. Combined with its outstanding performance in linear settings (reported in Section 4.1), one may find that the variational autoencoder based

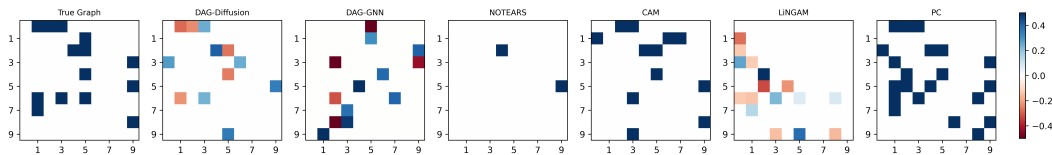

Figure 2: Estimations of the graph on a nonlinear model with Gaussian noise (10 nodes).

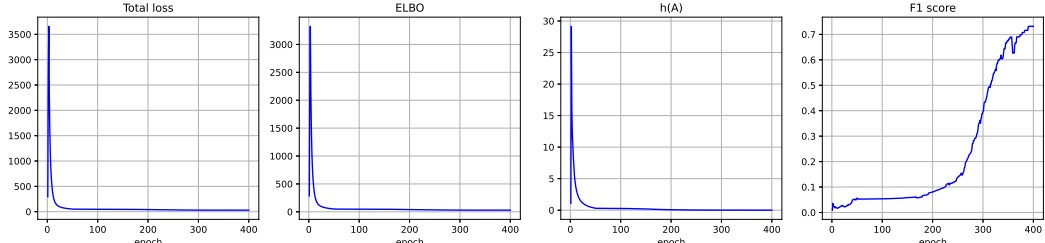

Figure 3: Learning process of DAG-Diffusion on a nonlinear model with Gaussian noise (50 nodes).

generative modeling for causal discovery seems to be better at dealing with simpler causal functions, although the model assumption considers nonlinear ones. However, DAG-Diffusion is possibly a good choice facing nonlinear models. This is evidenced by its more obvious empirical performance advantage compared to linear cases.

We visualize one randomly picked example of causal graphs discovered by different algorithms in 10-node case in Figure 2. One can see that DAG-GNN and NOTEARS generate sparse predictions, while PC tends to output relatively dense ones. LiNGAM generates a lot of wrong predictions. Although the edge predictions of our algorithm are not always correct, the results are closest to the given ground truth compared to others. An example learning curve is also reported in Figure 3. Initially, the loss value has a drastic increment due to the fact that the direction of optimization is close to random guess. The generated graph is consequently distant from the ground truth, and is cyclic evidenced by the high $h(A)$ shown in the figure. Along with training procedure, the loss value and $h(A)$ decrease, with the adjacency matrix updated to be with higher similarity to the true graph. Another observation is that although the single-step loss reduction becomes relatively minor compared to its initial state, the F1 score progressively increases along with the training procedure.

### 4.3 REAL DATA

We use two real world datasets for algorithm evaluations. The first is the protein signaling network, constructed by the data of different expression levels of proteins, Sachs (Sachs et al., 2005). The annotations are widely accepted by the community, with edges at least partially verified by biological experiments. The datasets contain both observational and interventional data, and we use the observational data with totally 863 samples. The ground truth graph (Sachs et al., 2005) has 11 nodes with 17 edges.

The true graph is sparse so that an empty graph has a SHD as 17, and the causal relations are probably highly nonlinear due to the complex nature of biological systems. Instead of reporting the FDR as those in previous sections, we record Number of Non-Zeros (NNZ) in the estimated adjacency matrix to show the number of edges in the outputs. The empirical results are reported on Table 5. We observe that PC and GraN-DAG output some edges that do not exist in the true graph, resulting in unsatisfactory performance in terms of SHD. Our method, as well as DAG-GNN show some promising empirical records, compared to other approaches, providing partial evidence on these methods' capability on nonlinear modeling. It is worth a discussion that NOTEARS also achieves the best SHD. However, compared to the results of our approach, the smaller TPR indicates that the graph generated by NOTEARS is relatively sparse with some undiscovered edges. It is a bit surprising that NOTEARS-MLP gets the worst performance among all methods. Although the inclusion of MLP modules to approximate the nonlinear functions makes the method theoretically applicable to

Table 5: **Empirical results on Sachs dataset.**

|  | DAG-Diffusion | PC | CAM | ICA-LiNGAM | DAG-GNN | GraN-DAG | NOTEARS | NOTEARS-MLP |
|---|---|---|---|---|---|---|---|---|
| TPR | 0.47 | 0.41 | 0.35 | 0.17 | 0.58 | 0.23 | 0.35 | 0.47 |
| NNZ | 16 | 13 | 10 | 8 | 20 | 21 | 14 | 43 |
| SHD | 12 | 22 | 12 | 15 | 14 | 27 | 12 | 36 |
| F1 score | 0.47 | 0.24 | 0.42 | 0.15 | 0.54 | 0.09 | 0.3 | 0.17 |

Table 6: **Empirical results on SynTReN (20 nodes) dataset**.

|  | DAG-Diffusion | PC | CAM | ICA-LiNGAM | DAG-GNN | GraN-DAG | NOTEARS | NOTEARS-MLP |
|---|---|---|---|---|---|---|---|---|
| TPR | 0.5 | 0.16 | 0.2 | 0.29 | 0.125 | 0 | 0.54 | 0.29 |
| NNZ | 35 | 48 | 36 | 88 | 20 | 21 | 76 | 115 |
| SHD | 33 | 57 | 45 | 90 | 43 | 43 | 70 | 115 |
| F1 score | 0.34 | 0.027 | 0.05 | 0.04 | 0.03 | 0 | 0.18 | 0.03 |

nonlinear cases, the actual performance in Sachs is worse than its linear version. The CAM's performance is satisfactory, considering the hardness of the data. This indicates the CAM's ability in dealing with complex situations with noise distributions unknown and possibly heterogeneous.

The results on another dataset SynTReN (Van den Bulcke et al., 2006) are reported in Table 6. It is a synthetic transcriptional regulatory network generator that can produce synthetic gene expression data to approximate experimental data. The network topology is sampled from previously known regulatory networks. In this regard, the data can also be treated as pseudo real data due to some randomness caused by artificial sampling. In the experiments, we follow the configurations of data generation used by Lachapelle et al. (2019) (setting probability for complex 2-regulator interactions to be 1 with random seed 0) and draw 500 samples from a graph with 20 nodes to construct the evaluation dataset.

In this data, the underlying distributions of variables are potentially quite complex due to the usually sophisticated interactions between genes in whole genomes. Our approach achieves the best results in terms of SHD among all compared approaches. It is noticeable that DAG-GNN tends to output sparse graphs in this case, evidenced by the small reported NNZ value. On the contrary, NOTEARS prefers a dense graph based on the principle of minimizing the reconstruction error under linear model assumptions. ICA-LiGNAM fails to predict most edges, possibly due to the heavy violation of their assumptions because of the inapproximability of underlying gene regulatory mechanisms by simple linear functions. Although NOTEARS-MLP has consistently good performance in synthetic data, it tends to output many edges in SynTReN network, and consequently with large recorded NNZ values. This shows that NOTEARS-MLP still has certain limitations in modeling causal relations that are of potentially high complexity.

## 5 CONCLUDING REMARKS

In this paper, we propose a novel approach for causal discovery utilizing diffusion models within a continuous optimization framework. We integrate a smooth characterization of the acyclicity constraint into our objective function to generate acyclic adjacency matrices. The diffusion process's prowess in nonlinear modeling provides some theoretical justifications for our model's capacity to uncover causal graphs from observational data stemming from possibly intricate generative mechanisms. Extensive experiments on linear and nonlinear models with various types of noise, as well as real world datasets, are conducted. The empirical results show that our proposed model achieves satisfactory accuracy in causal discovery.

For future work, it is remarkable that training diffusion models, particularly those with deep layers, remains a computational challenge. Improving the training efficiency of diffusion causal models is consequently a promising direction for future research. Furthermore, determination of the configuration parameters of diffusion process, such as its depth, remains largely heuristic. Research on techniques for automatic parameter tuning is thus important for its performance on various application scenarios.

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

## APPENDIX

## A    PROOF OF PROPOSITION 1

The equivalence condition between unconstrained optimization problem and its transformed constrained optimization problem has been discussed in some literature (Boyd & Vandenberghe, 2004). We follow a similar logic of (Zhu et al., 2019) to complete the proof. Suppose that $A$ is a solution to the problem (17). Then let $\mathcal{L}_* = \mathcal{L}(A, \Theta_*)$ with $\Theta_*$ being some coupled parameters for the minimization problem. Clearly, $A$ is acyclic due to the hard constraint in (17). If $A$ is not a solution to the problem (18), then this indicates that there exist an adjacency matrix $A'$ such that

$$\mathcal{L}_* > \min_{\Theta} \mathcal{L}(A', \Theta) + \lambda h(A'). \tag{24}$$

By assumption, $A'$ cannot be acyclic, otherwise we have a directed acyclic graph that corresponds to a score lower than the assumed minimum $\mathcal{L}_*$. Then it follows that

$$\min_{\Theta} \mathcal{L}(A', \Theta) + \lambda h(A') \geq \mathcal{L}_l + \lambda h_* \geq \mathcal{L}_u, \tag{25}$$

which violates the assumption that $\mathcal{L}_u$ upper-bounds the loss function.

Then we consider the other direction. Suppose there exists a solution $A$ that satisfies problem (18) but is not a solution to problem (17). Then there exist two cases: i) $A$ is not acyclic; ii) $A$ is acyclic but yields a higher loss value than the assumed minimum. The second case makes a clear contradiction of the definition of minimum loss. For the first case, assume that there is a acyclic matrix $A'$ that achieves the minimum score. Then one can easily get the inequality of (24) since $h(A') = 0$ in this case, which leads to a contradiction that $A$ minimizes the loss function.

## B    DETAILS ABOUT EXPERIMENTAL SETTINGS

The details of the experimental configurations are listed here.

- Diffusion process. Our settings of diffusion models vary across experiments on different graph sizes. The parameters of diffusion model include the diffusion step $T$, the initialization of maximum and minimum variances and the design of the denoising neural networks. The $(I - A)$ is placed on the medium layer of the forward process, and the reverse process is with a structure symmetrical to the forward process. The denoising network consists of multiple layers, with structure

    nn.Linear(64, 64)

    nn.LeakyReLU(negative slope = 0.1)

    Here, the variance sequence is initialized with (min, max) as $(e^{-4}, 0.2)$.

- Settings of MLP in (23). In the experiments on synthetic data, a MLP is used to simulate nonlinear causal models. For each dimension of data, we use 3 layers with each edges sampled randomly from normal distributions, with configurations below.

    self.fc1 = nn.Linear(1, 256)

    self.fc2 = nn.Linear(256, 256)

    self.fc3 = nn.Linear(256, 1)

    The variable passes through a $sin(X)$ function and a LeakyRelu with negative slope 0.3, except for the output layer where a $tanh$ is applied. Skipped connections as that used by ResNet (He et al., 2016) are adopted to add the input directly to the last layer.

- Updating rules of augmented Lagrangian. Recall the $i$th subproblem as

    $$\min_{A, \Theta} \mathcal{L}(A, \Theta) + \lambda_i h(A) + \frac{c_i}{2} h(A)^2. \tag{26}$$

    Parameter updating is performed after each subproblem resolved. Suppose the $A_i$ is the solution for subproblem $i$. Then the updating rules are

    $$\lambda_{i+1} = \lambda_t + c_i h(A_i),$$

    $$c_{i+1} = \begin{cases} \eta c_i, & \text{if } h(A_i) > \omega h(A_{i-1}), \\ c_i, & \text{otherwise.} \end{cases} \tag{27}$$

We set $(\eta, \omega) = (5, 0.05)$ for the experiments. Each subproblem is with an initial point inherited from the previous one $A_{i-1}$. The procedures are also recorded in Algorithm 1.

- Hyperparameter searching. Our method employes a hyperparameter search to optimize the performance, which is similar to that used by GraN-DAG method. In the hyperparameter search procedure, we select the combination of parameters by BIC score, in an unsupervised manner. There are several hyperparameters including learning rate, batch size, search times, hidden layers, hidden unit in denoising layers, length of forward process, and length of reverse process. The possible values of different parameters are listed in Table 7.

Table 7: Hyperparameter search space.

|  | **Hyperparameter space** |
|---|---|
| **DAG-Diffusion** | learning rate $\sim \mathcal{U}[5e{-}5, 1e{-}4, 5e{-}4]$
batch size $\sim \mathcal{U}[20, 50]$
search times $\sim \mathcal{U}[10, 15, 30, 50]$
number of hidden layers $\sim \mathcal{U}[1, 2, 3]$
number of hidden units in denoising layers $\sim \mathcal{U}[16, 32, 64]$
length of forward process $\in \{3, 4, \ldots, 10\}$
length of reverse process $\in \{3, 4, \ldots, 10\}$ |

## C  ALGORITHM IMPLEMENTATIONS

We use the exiting implementations of causal discovery algorithms for comparisons, as below:

- NOTEARS (Zheng et al., 2018) is a causal discovery algorithm targeting linear models. It recovers the causal graph by estimating a weighted adjacency matrix by optimization under the least square reconstruction loss, combined with a smooth characterization for the acyclicity constraint. Thresholding is applied on the estimated weights to produce the inferred directed acyclic graph. Codes are available at the GitHub repository `https://github.com/xunzheng/notears`.

- NOTEARS-MLP (Zheng et al., 2020) extends NOTEARS to nonlinear cases incorporating nonlinear regression functions. It takes use of neural networks to approximate arbitrary $f$. Codes are also available at the GitHub repository `https://github.com/xunzheng/notears`.

- DAG-GNN (Yu et al., 2019) is a causal discovery algorithm that formulates the problem within a variational autoencoder framework, utilizing graph neural networks as its encoder and decoder structure. With a modified acyclicity constraint, the weighted adjacency matrix is generated by optimizations with the evidence lower bound as its loss function. The Python code is available at the GitHub repository `https://github.com/fishmoon1234/DAG-GNN`.

- GAE (Ng et al., 2019) performs causal discovery via a graph auto-encoding framework. The causal graph is obtained by optimizing a specially designed encoder-decoder model. The implementation is available at the GitHub repository `https://github.com/huawei-noah/trustworthyAI/tree/master/gcastle/castle/algorithms/gradient/gae`.

- GraN-DAG (Lachapelle et al., 2019) is a causal discovery algorithm that employs feedforward neural networks to model causal relationships. The the $(i, j)$-th element of the weighted adjacency matrix is determined by considering the sum of all product paths between variables $x_i$ and $x_j$. Utilizing the acyclicity constraint introduced by NOTEARS (Zheng et al., 2018), GraN-DAG identifies a directed acyclic graph (DAG) by maximum likelihood estimations, over observed data. The implementation is available at the GitHub repository `https://github.com/kurowasan/GraN-DAG`.

- PC (Spirtes et al., 2001) employs greedy search originated from the Greedy Equivalence Search (GES) algorithm (Chickering, 2002). ICA-LiNGAM (Shimizu et al., 2006) is a causal discovery algorithm that assumes a linear non-Gaussian additive

model, and employs independent component analysis to recover the weighted adjacency matrix. CAM (Bühlmann et al., 2014) searches for causal ordering among variables by applying feature selections and edge selections, under an assumption of nonlinearity on causal functions. All the algorithms use pruning methods as post-processing steps. Codes of algorithms are accessible through the py-causal package, available at `https://github.com/FenTechSolutions/CausalDiscoveryToolbox`.

- DiffAN (Sanchez et al., 2022) is a topological ordering search algorithm under the framework of nonlinear additive noise model, with the Hessian matrix of data estimated by diffusion models. The code is available at `https://github.com/vios-s/DiffAN`.

## D    SOME EXPERIMENTS AND DISCUSSIONS

We firstly discuss the type of models the functional formulation in (11) can adopt. If $g$ and $f$ are identify functions, we get linear causal models such as LiNGAM. If $g$ is some nonlinear function and $f$ is an identity function, we get nonlinear additive noise causal models with some linear mixing mechanism (represented by mixing matrix $A$). More interestingly, let $f$ be some mixing function. We get additive noise model with noise post-processing, so that the "added" variables can be mutually non-independent.

We also record the performance of the method on models with different sample size in Figure 4.

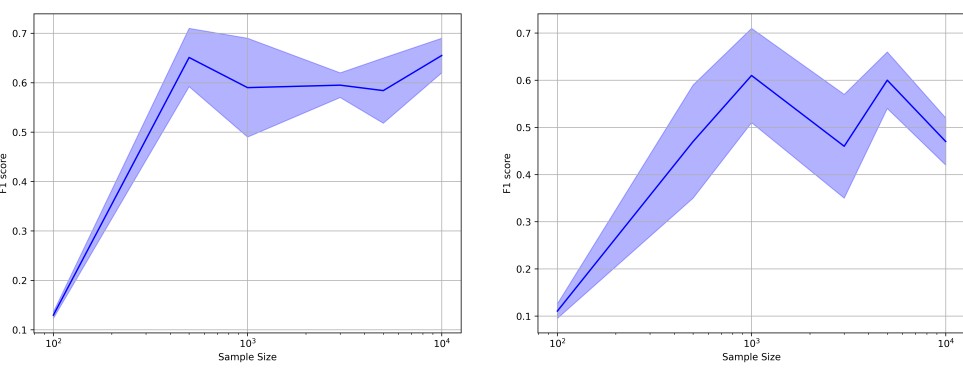

(a) Nonlinear model with Gaussian noise.          (b) Nonlinear model with non-Gaussian noise.

Figure 4: F1 score versus sample size on different models.