# OpenReview forum: "Gradient based Causal Discovery with Diffusion Model"
_ICLR.cc/2025/Conference — Submitted to ICLR 2025_

### Official Review · Reviewer_CBSw · 2024-10-28

**Soundness:** 2
**Presentation:** 3
**Contribution:** 2
**Rating:** 5
**Confidence:** 3

**Summary:**

This paper introduces a diffusion model (denoised diffusion probabilistic model) for causal discovery and demonstrates its effectiveness in this task.

**Strengths:**

The paper highlights the significant advantages of using a diffusion model for causal discovery. In nonlinear models, it consistently outperforms many other baseline methods.

**Weaknesses:**

* Small typo: Line 309-310, "deep leyers"-> "deep layers". Line 183-184, "node i to node j"->"node $i$ to node $j$"


* The current form of Proposition 1 lacks rigor, raising concerns about its accuracy and possible need for revision. Here, $A$ is the weighted adjacency matrix. Notably, a sequence $A_n$ can always be constructed such that $\inf_{n} h(A_n) = 0$ (using $\inf$ here is more appropriate than $\min$), as $A$ is a weighted adjacency matrix and $h$ is continuous in $A$, implying that $h_* = 0$. Additionally, the phrase “all direct cyclic graphs in the solution space” is unclear and seems incorrect. The solution space should consist of only directed acyclic graphs, so there should be no cyclic graphs in this space. Please revise this accordingly.

* The experimental results omit the algorithm’s runtime, and results are only reported for graphs with up to 50 nodes and for ER2. Given that this paper employs a diffusion model for causal discovery, it would be helpful to include [1] as a baseline. Additional experiments would strengthen the case for the effectiveness of the diffusion model in causal discovery.

[1]Pedro Sanchez, Xiao Liu, Alison Q O’Neil, and Sotirios A Tsaftaris. Diffusion models for causal discovery via topological ordering. arXiv preprint arXiv:2210.06201, 2022.

**Questions:**

* I noticed that CAM predicts many edges that do not exist in the true causal graphs. This could be due to the CausalDiscovery package having the tuning step turned off by default, which is crucial for the CAM algorithm. Could this be a contributing factor to CAM’s degraded performance?

* Could you clarify why DAG-diffusion’s performance is lower in linear models? Is it possibly because the diffusion model is more effective at capturing nonlinear relationships?

---

> ### Author Response · Authors · 2024-11-23
> **Responses to Reviewer CBSw**
>
> Thank you for the comments. Here are our revisions based on your advises.
>
> &nbsp;
>
> Weaknesses 1:Small typo: Line 309-310, "deep leyers"-> "deep layers". Line 183-184, "node i to node j"->"node $i$ to node $j$"
>
> A: Thank you. We fixed this in the updated version.
>
> &nbsp;
>
> Weaknesses 2:The current form of Proposition 1 lacks rigor, raising concerns about its accuracy and possible need for revision. Here, $A$ is the weighted adjacency matrix. Notably, a sequence $A_n$ can always be constructed such that $\inf_{n}h(A_n)=0$ (using inf here is more appropriate than $\min$), as $A$ is a weighted adjacency matrix and $h$ is continuous in $A$, implying that $h_∗=0$. Additionally, the phrase “all direct cyclic graphs in the solution space” is unclear and seems incorrect. The solution space should consist of only directed acyclic graphs, so there should be no cyclic graphs in this space. Please revise this accordingly.
>
> A:Thank you for raising this concern. For the solution space, we here mean all possible matrices in the search space, or the matrices in the optimization trajectory, not solution space. We change this in the updated version.
>
> &nbsp;
>
> Weaknesses 3:The experimental results omit the algorithm’s runtime, and results are only reported for graphs with up to 50 nodes and for ER2. Given that this paper employs a diffusion model for causal discovery, it would be helpful to include [1] as a baseline. Additional experiments would strengthen the case for the effectiveness of the diffusion model in causal discovery.
>
> A: This is a good advise. We include this as a baseline in the updated paper. The main consideration of this paper is the capability of the proposed causal method for functional causal discovery. We do not really pay much attention to running time, but care more about accuracy.
>
> &nbsp;
>
> Q1: I noticed that CAM predicts many edges that do not exist in the true causal graphs. This could be due to the CausalDiscovery package having the tuning step turned off by default, which is crucial for the CAM algorithm. Could this be a contributing factor to CAM’s degraded performance?
>
> A:Thank you for this advice. We change the parameter and report the results in the updated version.
>
> &nbsp;
>
> Q2: Could you clarify why DAG-diffusion’s performance is lower in linear models? Is it possibly because the diffusion model is more effective at capturing nonlinear relationships?
>
> A: We agree with your arguments, and we make several comments on this in the experiment section (see line 453-454).

---

> > ### Comment · Reviewer_CBSw · 2024-11-29
> >
> > Thank you for addressing my concern about the paper.
> >
> > However, I believe there is an issue with Proposition 1. Consider the following simple counterexample:
> >
> > Let
> > $$
> > A_n = \begin{pmatrix} \frac{1}{n} & 0 \\\\ 0 & 0 \end{pmatrix}.
> > $$
> >
> > Here, we have $ h(A_n) = \mathrm{Tr}(\exp(A_n \circ A_n)) - 2 = \exp\left(\frac{1}{n^2}\right) - 1 $. Since $A_n$ represents a directed cyclic graph, it is evident that as $ n \to \infty $, $ h(A_n) \to 0 $. In this case, $ h_* = 0 $.
> >
> > This example demonstrates that your conclusion is not generally correct.

---

> > > ### Author Response · Authors · 2024-11-30
> > > **Responses to Reviewer CBSw**
> > >
> > > Thank you for the response. This is a good consideration.  In fact, this graph you give is very special. What we mean "weighed adjacency graph" is not one that have edge strength on the diagonal element that converge to 0, because this means that we have a graph in infinity case that have almost no edge at all, which is acylic, not cyclic. The more easy-to-understand case is we treat this as binary matrix, where the $\frac{1}{n}$ is replaced with $1$ so that this is a binary adjacency matrix. Then your calculation will give you another result.  The more general insight is that (equation (5) in reference Zheng 2018) interpreting $h(A)$ as $tr(I-A)^{-1}=tr\sum_{k=0}^\infty A^k=n+tr\sum_{k=1}^\infty A^k$, where $trA^k$ counts the number of length-k closed walks in a directed graph, and must be 0 if acyclic. In this regard, cyclic graph will not lead to an $h_*$ being 0 even in the infinity case, and it would be more clear to interpret the case in a finite dimensional space to remove ambiguity.
> > >
> > >
> > > If you have further questions, please kindly let us know and we are happy to make discussions to make issues resolved.
> > >
> > >
> > >
> > > Reference: Zheng X, Aragam B, Ravikumar P K, et al. Dags with no tears: Continuous optimization for structure learning. Advances in neural information processing systems, 2018, 31.

---

### Official Review · Reviewer_eoGv · 2024-11-03

**Soundness:** 3
**Presentation:** 3
**Contribution:** 2
**Rating:** 5
**Confidence:** 4

**Summary:**

This paper proposed an approach to combine the causal discovery with the diffusion model. In particular, the author proposed to adopt a DAG-GNN approach for structural causal model but use diffusion approach to model two of the non-linear function $f$, $g$. For DAGness of the graph, they propose to use NOTEARS approach for soft but differentiable constraints. Empirically, the author evaluates on 4 synthetic datasets and two semi-synthetic ones to evaluate the performance, showing that DAG-Diffuser can outperform the baselines on some datasets.

**Strengths:**

This paper is clearly written and easy to follow. The proposed approach is indeed sound and correct. Personally, although I was expecting the proposed method is to use diffusion for graph modelling (i.e. diffusion model to directly model the graph distribution), it is still interesting to see the performance if one uses diffusion model only for function modelling.

**Weaknesses:**

However, I have several concerns regarding this paper. First, methodology-wise, I think the contribution is not significant. The core idea of this paper is to use diffusion model to model function $f$, $g$. Due to the generality of diffusion model (i.e. not much technical requirements if one wants to use diffusion model), it does not pose significant technical challenges of simply plug-and-play. Therefore, in terms of methodology, it is of little difference compared to using normalizing flow, invertible neural network for $f$ and $g$, which has been done before.

Another question I have is the motivation of using diffusion model for $f$ and $g$. Is it because diffusion is flexible, and the author think that this may be helpful? If that is the case, another set of causal inference experiments can provide stronger evidence of choosing diffusion model. Since the proposed method in fact learns a structural causal model. it is straightforward to extend the framework for average treatment effect and individual treatment effect on synthetic data. Under this case, DAG-Diffuser may demonstrate stronger performance compared to baselines.

In addition, for empirical evaluation on linear Gaussian synthetic dataset, do you use the same variance for the Gaussian noise variable? If not, the model is not identifiable, and invalidate your statements.

Some minor comments:
1. For eq.2, is the $Y_0$ and $Y_T$ are in wrong order?

2. You should add some reference on line192. For example, [1].

3. In line 196, you mentioned that "without loss of generality, we assume....", but the invertibility of $f$ and $g$ will hurt the capacity of $f,g$.

4. For line 201, if f is non-linear, do you still need $Z$ to be non-Gaussian?

5. Line 261, what is $S_G$? Shouldn't it be $R_G$?

6. For related work, you should also cite some related work on discovery with soft constraints and diffusion model, like [2,3,4,5].

7. Consider introduce the method name DAG-Diffuser before the experiment section.

8. what is $(10\alpha)/\alpha$ in line 403?


Reference

[1] Hoyer, Patrik, et al. "Nonlinear causal discovery with additive noise models." Advances in neural information processing systems 21 (2008).

[2] Rolland, Paul, et al. "Score matching enables causal discovery of nonlinear additive noise models." International Conference on Machine Learning. PMLR, 2022.

[3] Lachapelle, Sébastien, et al. "Gradient-based neural dag learning." arXiv preprint arXiv:1906.02226 (2019).

[4] Geffner, Tomas, et al. "Deep end-to-end causal inference." arXiv preprint arXiv:2202.02195 (2022).

[5] Sanchez, Pedro, et al. "Diffusion models for causal discovery via topological ordering." arXiv preprint arXiv:2210.06201 (2022).

**Questions:**

See above

---

> ### Author Response · Authors · 2024-11-23
> **Responses to Reviewer eoGv**
>
> Thank you for the comments. Here are our revisions based on your advises.
>
> &nbsp;
>
> Weakness 1: "However, I have several concerns ... done before"
>
> A: This is an important comment. The main difference between normalizing flow and diffusion model is the invertibility of the function and the estimation approach (consequently the complexity of the distribution that can be represented). The diffusion model has the flexibility of configuring the "noise adding" process while normalizing flow relies on the Jacobian matrix of functions to perform model estimation. Although we "assume" invertibility for the ease of explanation, our model works as long as the data obeys a generative process described by equation (1), without the actual need to assume invertibility. This differs from methods using NF, with more flexibility to represent observational distributions. In this regard, this framework using the diffusion model as a tool contributes an important piece to the actual discovery methods, which is with wider applicability, and we think is not so simply a "plug-in". We also change the statements of invertibility in the main text (kindly see rebuttal to your Q3).
>
> Reference
> https://deeplearning.cs.cmu.edu/F23/document/slides/lec23.diffusion.updated.pdf
>
> &nbsp;
>
> Weakness 2: "Another question I have is the motivation ... compared to baselines."
>
> A: Thank you for this comment. This is indeed an good idea and a direct extension in fact. Since our paper mainly focuses on causal discovery, we do not compare the approach with causal inference ones. But this is a good and important extension, we may list as future work.
>
> &nbsp;
>
> Weakness 3: "In addition, ... and invalidate your statements."
>
> A: Yes, we use equal variance in the experiments.
>
> &nbsp;
>
> Q1: For eq.2, is the $Y_0$ and $Y_T$ are in wrong order?
>
> A:The forward process corresponds to the process transforming $Y_0$ to $Y_T$, and the reverse process corresponds to the process that reverses this chain. The order in eq.2 simulates the chain process in diffusion models.
>
> &nbsp;
>
> Q2:You should add some reference on line192. For example, [1].
>
> A:Thank you for this issue. We made modifications in the new version. (see line 192)
>
> &nbsp;
>
> Q3: In line 196, you mentioned that "without loss of generality, we assume....", but the invertibility of $f$ and $g$ will hurt the capcity of $f$,$g$.
>
> A: Thank you for pointing out this. We modify this to "when $f$ and $g$ are invertible, we can write " to remove ambiguity. This also closely relates to your concerns raised in weakness section.
>
> &nbsp;
>
> Q4: For line 201, if f is non-linear, do you still need $Z$ to be non-Gaussian?
>
> A: This relates to identifiability of the model, while in our model in fact we do not consider too much on this. The variable $Z$ here are just exogenous variables.
>
> &nbsp;
>
> Q5:Line 261, what is $S_G$? Shouldn't it be $R_G$?
>
> A: Thank you for this. We fixed it.
>
> &nbsp;
>
> Q6: For related work, you should also cite some related work on discovery with soft constraints and diffusion model, like [2,3,4,5].
>
> A:Some of them have been mentioned in other content, and [3,5] have been used as baselines. As for [2,4], we discuss them in related work.
>
> &nbsp;
>
> Q7: Consider introduce the method name DAG-Diffuser before the experiment section.
>
> A: We make this modifications as you instructed.
>
> &nbsp;
>
> Q8: What is $(10α)/α$ in line 403?
>
> A:Sorry for this typo. It‘s $(10α)/10$. This is a kind of method to choose an appropriate threshold.
>
>
> &nbsp;
>
>
> Reference
>
> [1] Hoyer, Patrik, et al. "Nonlinear causal discovery with additive noise models." Advances in neural information processing systems 21 (2008).
>
> [2] Rolland, Paul, et al. "Score matching enables causal discovery of nonlinear additive noise models." International Conference on Machine Learning. PMLR, 2022.
>
> [3] Lachapelle, Sébastien, et al. "Gradient-based neural dag learning." arXiv preprint arXiv:1906.02226 (2019).
>
> [4] Geffner, Tomas, et al. "Deep end-to-end causal inference." arXiv preprint arXiv:2202.02195 (2022).
>
> [5] Sanchez, Pedro, et al. "Diffusion models for causal discovery via topological ordering." arXiv preprint arXiv:2210.06201 (2022).

---

> ### Comment · Reviewer_eoGv · 2024-11-28
>
> Thanks for author's response, my argument is not about the similarity of NF and diffusion model-wise, but refer to little difference methodology-wise. If one uses NF for $f$ and $g$ (which is a straight-forward modification), it does not require significant modifications if one wants to replace it with diffusion model.

---

> ### Author Response · Authors · 2024-11-29
> **Responses to Reviewer eoGv**
>
> Thank you for the responses. In fact, concerning the main difference between the two methods, you already give very good comments: the invertibility of the functions. In this regard, our model is able to represent richer conditional distributions when admitting non-invertible functions in (1). The distributions are also discussed in equation (15). We already make modifications of statements in the updated pdf, and we give our sincere thanks for your advice which greatly improves the quality of the paper in terms of its theoretical perspectives. Although it seems very direct to replace diffusion with NF, the underlying estimation approaches (NF with Jacobian and DF with likelihood) and consequently the “richness” of representable conditional distributions are obviously different. From this perspective, the seemingly “simple” replacement makes some difference, and contributes to society with a piece of useful tool for causal discovery task.

---

### Official Review · Reviewer_6nj7 · 2024-11-04

**Soundness:** 2
**Presentation:** 2
**Contribution:** 1
**Rating:** 5
**Confidence:** 3

**Summary:**

This paper adopts diffusion models for differentiable causal discovery. Specifically, a specific function class is considered, where diffusion models are used to model the nonlinear causal relations. Empirical studies on synthetic and real data are provided.

**Strengths:**

Causal discovery with nonlinear relations is an important task, because it is often unrealistic to assume linear relations in practice.

**Weaknesses:**

- The specific nonlinear functional class considered/assumed is highly restrictive. The method cannot handle more general functional causal model, such as nonlinear additive noise models.
- The baselines considered are not adequate.

**Questions:**

- I would suggest the authors to be clear about what type of functional causal model the method can handle, which is not clear from Eq. (10), (11), (12). Specifically, the paper should give a precise formulation in the form of a structural causal model $X=f_i(PA(X_i),Z_i)$.
- Is $f$ and $g$ in Eq. (10) and (11) a variable/element wise function? If so, this should be stated explicitly.
- Only CAM and DAG-GNN are compared for nonlinear models. Several other differentiable nonlinear methods could be included to strengthen the empirical studies, such as those for more general nonlinear causal models (https://arxiv.org/abs/1909.13189, https://arxiv.org/abs/1906.02226), and those for the specific causal model in Eq. (11) (https://arxiv.org/abs/1911.07420, https://arxiv.org/abs/2004.08697)

Minor:
- Proposition 1 follows similar spirit as Zhu et al. (2020), which should be made more explicitly in the main paper. Also, Proposition 1 does not provide any new insight into the optimization procedure because a typical way in differentiable causal discovery is to use augmented Lagrangian, which the paper does, so I would suggest removing this proposition.
- L56: Zheng et al. (2020) did not propose a nonparametric score function but a nonparametric DAG constraint.

---

> ### Author Response · Authors · 2024-11-23
> **Responses to Reviewer 6nj7**
>
> Thank you for the comments. Here are our revisions based on your advises.
>
> Weakness 1: The specific nonlinear functional class considered/assumed is highly restrictive. The method cannot handle more general functional causal model, such as nonlinear additive noise models.
>
> A: We thank the reviewer for pointing out this issue. In fact, the nonlinear functional class is not restrictive, but this form admits a wide class of models. By changing the functions in the equation (11), we can get a lot of "traditional" models
> 1.	Set $g$ and $f$ to be identify function, we get linear causal models
> 2.	Set $g$ to be some nonlinear function and f to be identity function, we get nonlinear additive noise causal models with some linear mixing mechanism (represented by mixing matrix A).
> 3.	Set $f$ to be a mixing function, we get additive noise model with noise post-processing, so that the "added" noise can be mutually non-independent.
> We thus think this model is not restrictive. A discussion in added in Appendix D.
>
> &nbsp;
>
> Q1: I would suggest the authors to be clear about what type of functional causal model the method can handle, which is not clear from Eq. (10), (11), (12). Specifically, the paper should give a precise formulation in the form of a structural causal model.
>
> A：Thank you for raising this issue. Basically, our model originated from the linear causal model (eq (8)) and then extends to nonlinear cases, and formulating this into a structural causal model does not naturally match our explanation why diffusion process can model the generative functions. We thus prefer to state the model under the functional generative model class. To make "what type of functional causal model the method can handle" more clear, we put down additional discussions in appendix (also summarized in our response to your weakness one).
>
> &nbsp;
>
> Q2:Is $f$ and $g$ in Eq. (10) and (11) a variable/element wise function? If so, this should be stated explicitly.
>
> A: Here we consider $f$ and $g$ to be variable wise function. We also state this in the updated version (between line 191 to line 200).
>
> &nbsp;
>
> Weakness 2: The baselines considered are not adequate.
>
> Q3: Only CAM and DAG-GNN are compared for nonlinear models. Several other differentiable nonlinear methods could be included to strengthen the empirical studies, such as those for more general nonlinear causal models (https://arxiv.org/abs/1909.13189, https://arxiv.org/abs/1906.02226), and those for the specific causal model in Eq. (11) (https://arxiv.org/abs/1911.07420, https://arxiv.org/abs/2004.08697)
>
> A: Thank you for mentioning more related work. We respond to your comments one by one.
>
> 1.	GraN. We compared this in the real world dataset.
>
> 2.	GAE and NOTEARS-MLP. We added the experiments in tables. Please see the updated version.
>
> 3.	CausalVAE. This method needs labels as supervised signals, and mainly targets images for hidden concept learning. It is not direct applicable in our testing dataset. However, this is indeed related work and we discuss it in section 2.
>
> &nbsp;
>
> Minor:
> Proposition 1 follows similar spirit as Zhu et al. (2020), which should be made more explicitly in the main paper. Also, Proposition 1 does not provide any new insight into the optimization procedure because a typical way in differentiable causal discovery is to use augmented Lagrangian, which the paper does, so I would suggest removing this proposition.
> L56: Zheng et al. (2020) did not propose a nonparametric score function but a nonparametric DAG constraint.
>
> A: Thank you for this comment. The proposition we still keeps because it states the euqvalence between augmented optimization procedures which is important property and can make our paper self-contained. We address your comment "more explicitly" (see line 285) and "a nonparametric score function but a nonparametric DAG constraint" by making modifications on this in the updated version. L56 is revised as "the score function with nonparametric DAG constraint". (see line 58)

---

> > ### Comment · Reviewer_6nj7 · 2024-11-25
> >
> > Thanks for the response. Some of my concerns have been addressed and I have updated my rating accordingly.

---

> > > ### Author Response · Authors · 2024-11-26
> > > **Responses to Reviewer 6nj7**
> > >
> > > Thank you for the response. If there are other questions or uncleared concerns, please kindly let us know and we are willing to provide more materials and revise our manuscripts accordingly.

---

### Official Review · Reviewer_FcqQ · 2024-11-07

**Soundness:** 2
**Presentation:** 3
**Contribution:** 2
**Rating:** 5
**Confidence:** 3

**Summary:**

The authors propose using diffusion models for causal discovery and searching for the DAG under continuous optimization frameworks.  The authors claim that the diffusion model has the ability to represent various functions, and the proposed causal discovery approach is able to generate graphs with satisfactory accuracy on observational data generated by either linear or nonlinear causal models.  Experiments on synthetic and real-world datasets were conducted to test the proposed method.

**Strengths:**

The authors present different experiments on synthetic and real-world datasets. The writing is clear and easy to follow.

**Weaknesses:**

a. The method proposed in the paper is incremental, and it heavily relies on NoTears regularization.  The method is just a combination of diffusion model and NoTears constraint. The contribution of the paper is limited.

b. The benefit of the method is limited. As shown in Table 1 and Table 2.

c. The abstract is too general and does not provide sufficient information to summarize the content of the article.  It should provide insights into the proposed method.

**Questions:**

See weaknesses.

---

> ### Author Response · Authors · 2024-11-23
> **Responses to Reviewer FcqQ**
>
> Thank you for the comments. Here are our revisions based on your advises.
>
> &nbsp;
>
> Qa:The method proposed in the paper is incremental, and it heavily relies on NoTears regularization. The method is just a combination of diffusion model and NoTears constraint. The contribution of the paper is limited.
>
> A：Thank you for this comment. The novelty of our method indeed lies on the capability of gradient based causal discovery on a more general nonlinear structure causal model, as shown in equation (1), compared to existing ones. The diffusion model is used to simulate complex nonlinear generative causal functions, which can be treated as a tool for causal reasoning under our proposed framework.
>
> &nbsp;
>
> Qb:The benefit of the method is limited. As shown in Table 1 and Table 2.
>
> A: Table 1 and 2 mainly record the performance on linear causal models. Since our main contribution is on nonlinear models, we think that comparable performance to SOTA on linear models does not restrict the main benefits of the method. In fact, this is also empirical evidence that our method is able to complete causal discovery tasks when the underlying model is linear.
>
> &nbsp;
>
> Qc:The abstract is too general and does not provide sufficient information to summarize the content of the article. It should provide insights into the proposed method.
>
> A:This is a good advice. The core insights are that we use diffusion process to simulate nonlinear causal generative functions, which admits the method to perform causal discovery on more complex situations with a richer class of observational distributions. We modified the abstract so that the insights are stated more clearly. Please see the updated version (see " The underlying nonlinear causal generative process is modeled with ...")

---

> > ### Comment · Reviewer_FcqQ · 2024-11-25
> >
> > Thank you for your response. I will keep the score unchanged.

---

> > > ### Author Response · Authors · 2024-11-26
> > > **Responses to Reviewer FcqQ**
> > >
> > > Thank you for the response. If there are other questions, please kindly let us know and we are willing to provide more materials and revise our manuscripts accordingly.

---

### Author Response · Authors · 2024-11-23
**Summary of revision**

Dear all,

We express our sincere thanks for the time you spent on our manuscript. We
revised our manuscript based on your comments, with the revisions summarized below.

1. Abstract:  abstract is revised so that the insight is more clear. (Reviewer FcqQ)

2. Section 2: more related work including GAE and CausalVAE, causal score matching are discussed. (Reviewer 6nj7, eoGv and CBSw)

3. Section 3: several revisions on statements related to invertibility of functions (line 197) and proposition 1. (Reviewer 6nj7, eoGv, CBSw)

4. Section 4: more experiments are added (GAE, DiffAN, NOTEARS-MLP) with discussions revised acordingly. Pruning method of CAM is also added. (Reviewer 6nj7, eoGv, CBSw)

5. Appendix: open source links (section C) and discussions (section D) are added about the type of nonlinear models and several new baselines. (Reviewer 6nj7)

For details, please kindly see our responses to each reviewer.

Authors

---

### Meta-Review · Area_Chair_X25v · 2024-12-21

**Metareview:**

The authors propose a diffusion-based approach to continuous causal discovery, using a DAG-GNN approach that models the non-linear function with diffusions. This is an interesting direction with a few prior works already. Reviewers unanimously rated the paper as "weak reject" with no one strongly in favor of acceptance.

**Additional Comments On Reviewer Discussion:**

Every reviewer engaged with the authors, with some increasing their score. But no one raised their score above "weak reject".

---

### Decision · Program_Chairs · 2025-01-22

Reject